# Study on the positivity rate and influencing factors of anxiety in pregnant women during the first fetal magnetic resonance examination: A cross-sectional study

**Yuping Zheng**[1,2], **Yun Wang**[1,2]*, **Xue Liu**[1,2], **Li Zhang**[1,2], **Hui Zhang**[1,2], **Juan Liu**[1,2], **Yang Liu**[1,2], **Xuesheng Li**[2,3], **Gang Ning**[2,3]

1 Department of Radiology Nursing, West China Second University Hospital, Sichuan University, Chengdu, China, 2 Key Laboratory of Birth Defects and Related Diseases of Women and Children (Sichuan University), Ministry of Education, Chengdu, Sichuan, China, 3 Department of Radiology, West China Second University Hospital, Sichuan University, Chengdu, China

* 532410227@qq.com

## Abstract

**Data Availability Statement:** All relevant data are within the manuscript.

### Objective

This study investigated the positive rate and related influencing factors of anxiety screening in pregnant women during the first fetal magnetic resonance examination.

### Methods

A total of 303 pregnant women who met the criteria for magnetic resonance pregnancy examination in a Grade III maternity hospital from December 2021 to December 2022 were included by the convenience sampling method. A cross-sectional survey was conducted before the examination using the General Situation Questionnaire and Self-rating Anxiety Scale (SAS).

### Results

The positive rate of anxiety was 31.02% (94/303), and the average score of anxiety was 45.71±9.84. Univariate analysis results showed that age, educational level, occupation, place of residence, per capita monthly income, and number of pregnancies were related to the anxiety status of pregnant women in the fetal magnetic resonance examination (*P*<0.05). The results of logistic regression analysis showed that the factor of college degree [OR: 2.168, 95% CI: (1.119, 4.273)] in the classification of cultural level and country factor [OR: 2.162, 95% CI: (1.066, 4.385)] in the classification of place of residence had an impact on the anxiety score of pregnant women in the fetal magnetic resonance examination (*P*<0.05).

**Funding:** The authors received no specific funding for this work.

**Competing interests:** The authors have declared that no competing interests exist.

## Conclusions

The positive rate of anxiety screening of pregnant women before the first prenatal magnetic resonance examination is high. A low education level and living in the countryside will increase the probability of anxiety in pregnant women during magnetic resonance examination. Based on the above research results, it is suggested that medical institutions pay attention to the mental health of pregnant women, improve mental health care services, and reduce the adverse psychological problems caused by prenatal examination.

## Introduction

Fetal magnetic resonance, as a supplementary imaging method to evaluate fetal abnormalities, can not only make a clear diagnosis of ultrasonic examination results, but also provide more diagnostic content without being limited by technical factors [1, 2], which has important prenatal diagnostic value. Foreign studies have shown that approximately 56.8% of pregnant women support magnetic resonance examination, but some women still express concerns about the impact of the examination on the fetus [3]. There is a temporary lack of research on the number of pregnant women with fetal magnetic resonance examinations in China, but previous similar studies have confirmed that fetal magnetic resonance examinations will increase the anxiety level of pregnant women and cause significant psychological distress [4]. Studies have shown that 5% to 10% of patients suffer from severe panic or claustrophobia during magnetic resonance examination [5]. In the psychological investigation of patients with magnetic resonance examination, the incidence of anxiety is as high as 45.2% [6]. Anxiety, as the main psychological reaction [7], has become an important predictor of the development of psychological problems in the process of magnetic resonance examination [8]. As a special group, pregnant women often have certain psychological pressure during the examination, which leads to physiological and psychological stress responses, such as fear, increased heart rate and elevated blood pressure [9]. Combined with the long time of examination, pregnant women show dizziness and tension, which increased the possibility of anxiety. Therefore, it is of great significance to pay attention to the anxiety of pregnant women during the fetal magnetic resonance examination.

The anxiety factors of pregnant women caused by magnetic resonance examination mainly come from the examination equipment and the subjects themselves [10]. On the one hand, the magnetic resonance examination space is closed. Due to the particularity of the examination, pregnant women need to be unable to move for a long time, and the internal environment is not comfortable. In addition, the noise generated by the equipment is large, and pregnant women lack the perception and control of the external environment, and tend to feel fear or anxiety [11]. On the other hand, women are more anxious than men in magnetic resonance examinations [12]. Due to different physiological mechanisms, physical fitness and social environments, women are more likely to show emotional behavior [13]. As a special group of women, pregnant women are worried about the impact of the examination on the fetus and the disease of the fetus. They are prone to conflict with the examination, and coupled with the strangeness of space when facing a separate examination [14], the anxiety problem is more prominent, which is not conducive to the smooth conduct of the examination [15, 16].

A meta-analysis shows that the detection rate of anxiety in prenatal pregnant women is 16% [17], and anxiety may lead to adverse maternal and infant outcomes. Studies have reported

that prenatal anxiety is associated with infants' white matter microstructure, which has a serious impact on early brain development [18]. At the same time, maternal anxiety causes fetal brain functional connectivity disorders, especially higher anxiety scores, which will affect the connection between the brainstem and the sensorimotor area, resulting in subsequent neurodevelopmental defects [19]. Meta-studies have shown that maternal anxiety has adverse birth outcomes with a risk of preterm birth and low birth weight, threatening the health of infants [20]. In addition, anxiety during pregnancy can also lead to postpartum hemorrhage, fetal distress and infection, which is not conducive to the health of pregnant women and fetuses [21]. It is very important to screen the mental health status of pregnant women with prenatal fetal magnetic resonance examination in a timely manner and put forward corresponding improvement measures according to the screening results. Therefore, on the basis of previous studies, this study conducted a cross-sectional survey of pregnant women undergoing the first fetal magnetic resonance examination through questionnaires to explore the anxiety status and influencing factors, aiming to reduce the anxiety level of pregnant women during fetal magnetic resonance examination, improve the comfort of examination, and provide an effective basis for improving the mental health of pregnant women.

## Materials and methods

In this study, a general situation questionnaire and self-rating anxiety scale were used to conduct a questionnaire survey on 303 pregnant women who underwent magnetic resonance pregnancy examination and met the criteria in the radiology clinic of a tertiary maternal and child specialist hospital in Chengdu. The survey time was from December 2021 to December 2022. This study was approved by the Ethics Committee of West China Second Hospital of Sichuan University. When filling in the questionnaire, the verbal consent of the subjects was obtained before the complete questionnaire was completed and complete data were finally obtained. If the subjects did not agree, we did not send them any questionnaires. The inclusion criteria were as follows: ①Pregnant women who participated in fetal magnetic resonance examination for the first time; ②Pregnant women who needed magnetic resonance examination due to abnormal fetal ultrasound results; ③Participants who had clear consciousness, and were able to communicate effectively, understood correctly and completed the questionnaire independently; ④Voluntary participation in the study. The exclusion criteria was pregnant women with severe mental illness. According to the requirements of statistical variable analysis, the sample size is calculated to be at least $5 \sim 10$ times the number of variables [22]. There were 14 predictors in this study, and the sample size was at least $70 \sim 140$ cases. Considering that there are approximately 20% invalid samples, the final sample is initially set at $84 \sim 168$ cases.

### General situation questionnaire

The questionnaire included pregnant women's age, gestational weeks, educational level, occupation, place of residence (it refers to the place where pregnant women live for a long time, which is divided into three categories in this article: country, town and city), living situation, per capita monthly income, number of pregnancies, number of births, number of abortions, attention to the fetus, understanding of magnetic resonance examination, risk status of magnetic resonance examination, and demand accompanied by family members.

### Self-rating Anxiety Scale (SAS)

The SAS was developed by Zung [23], with a total of 20 items and a total score of $20 \sim 80$ points. A 4-level scoring method was adopted, including 15 positive items and 5 reverse items,

which were scored according to the scoring order of 1–4 and 4–1 respectively. The standard score was rounded after the sum of the scores of each item was multiplied by 1.25. A standard score ≥50 indicates anxiety.

## Survey methods

A survey team was formed in advance and carried out the division of labor. After obtaining the consent of the subjects, the questionnaire was completed in the radiology outpatient department. The investigators adopted unified and standardized guidelines to ensure that each subject accurately understood the research content.

## Quality control

The investigation team was trained and qualified in advance. After repeated pre-investigation of all the questionnaire items, the questions were discussed and unified and standardized guidance was finally formed. After completing the questionnaire, the investigators checked whether there were any omissions in the questionnaire one by one and asked the subjects again to supplement and improve the unclear or missing points. After data collection, two researchers organized and summarized the data into an EXCELL form and double checked them to ensure the accuracy of data entry.

## Statistical analyses

SPSS 22.0 statistical software was used, frequency and percentage were used for statistical description of counting data, mean ± standard deviation was used for statistical description of measurement data, and independent sample $T$ test and variance analysis were used for statistical analysis. The influencing factors of anxiety were analyzed by logistic regression analysis. The test level was $\alpha = 0.05$, and $P<0.05$ indicated that the difference was statistically significant.

## Results

A total of 303 pregnant women who underwent fetal magnetic resonance examination were included in this study, with an average age of 29.92±4.09 years and an average gestational week of 28.59±4.15 weeks. The remaining data are shown in Table 1.

The positive rate of anxiety screening in this study was 31.02% (94/303), and the anxiety score was (45.71±9.84) scores. The univariate analysis results are shown in Table 1. The factors with statistical significance ($P<0.05$) in the univariate analysis were used as independent variables, and the anxiety score was used as the dependent variable (0 = without anxiety, 1 = with anxiety) for logistic regression analysis. The variables entering the regression equation were age (<35 years old = 0, ≥35 years old = 1), educational level (setting dummy variables with bachelor degree or above as reference), occupation (setting dummy variables with unemployed worker as reference), place of residence (setting dummy variables with city as reference), per capita monthly income (setting dummy variables with >5000 as reference), and number of pregnancies (setting dummy variables with more than three pregnancies as reference). Detailed data are shown in Table 2.

## Discussion

The main aim of this study was to explore the positive rate of anxiety in pregnant women who underwent fetal magnetic resonance examination for the first time and the influencing factors leading to anxiety. According to the literature review, there is a lack of research on anxiety in

**Table 1. The anxiety score of the pregnant woman in the fetal magnetic resonance examination.**

| Variable | | N(%) | $\bar{x} \pm s$ | F/t | P value |
|---|---|---|---|---|---|
| Age | <35 | 257 (84.8%) | 45.04±9.00 | -2.823 | 0.005* |
| | ≥35 | 46 (15.2%) | 49.43±13.11 | | |
| Gestational Weeks | <28 weeks | 127 (41.9%) | 45.29±10.01 | -0.623 | 0.534 |
| | ≥28 weeks | 176 (58.1%) | 46.01±9.73 | | |
| Educational Level | High school degree or below | 60 (19.8%) | 48.45±8.38 | 8.546 | <0.001* |
| | college degree | 100 (33.0%) | 47.47±10.19 | | |
| | bachelor degree or above | 143 (47.2%) | 43.32±9.65 | | |
| Occupation | manual worker | 81 (26.7%) | 47.81±11.49 | 4.832 | 0.009* |
| | mental worker | 157 (51.8%) | 44.05±9.06 | | |
| | Unemployed worker | 65 (21.5%) | 47.08±8.80 | | |
| Place of Residence | country | 68 (22.4%) | 49.53±11.11 | 7.995 | <0.001* |
| | town | 89 (29.4%) | 45.78±9.64 | | |
| | city | 146 (48.2%) | 43.88±8.83 | | |
| Living Situation | Couples live together | 152 (50.2%) | 45.51±9.80 | 1.962 | 0.120 |
| | Living with parents | 76 (25.1%) | 44.03±9.97 | | |
| | Living with parents-in-law | 66 (21.8%) | 47.95±9.97 | | |
| | Other ways of living | 9 (3.0%) | 46.78±5.54 | | |
| Per Capita Monthly Income | <1000 | 31 (10.2%) | 46.74±8.29 | 4.541 | 0.004* |
| | 1000∼2999 | 46 (15.2%) | 48.91±12.52 | | |
| | 3000∼4999 | 102 (33.7%) | 46.75±9.87 | | |
| | >5000 | 124 (40.9%) | 43.40±8.55 | | |
| Number of Pregnancies | 1 | 122 (40.3%) | 43.72±8.70 | 3.465 | 0.017* |
| | 2 | 95 (31.4%) | 47.57±11.75 | | |
| | 3 | 52 (17.2%) | 45.50±8.60 | | |
| | More than 3 times | 34 (11.2%) | 47.94±8.44 | | |
| Number of Births | 0 | 203 (67.0%) | 45.29±9.79 | 0.574 | 0.564 |
| | 1 | 93 (30.7%) | 46.61±9.77 | | |
| | 2 times and more | 7 (2.3%) | 45.71±12.70 | | |
| Number of Abortions | 0 | 176 (58.1%) | 44.57±9.53 | 2.845 | 0.060 |
| | 1 | 83 (27.4%) | 47.31±11.21 | | |
| | 2 times and more | 44 (14.5%) | 47.23±7.64 | | |
| Attention to The Fetus | NO | 2 (0.7%) | 45.00±9.90 | -0.102 | 0.919 |
| | YES | 301 (99.3%) | 45.71±9.86 | | |
| Understanding of Magnetic Resonance Examination | NO | 224 (73.9%) | 45.86±9.60 | 0.462 | 0.644 |
| | YES | 79 (26.1%) | 45.27±10.55 | | |
| Magnetic resonance examination is risky | NO | 199 (65.7%) | 45.42±9.93 | -0.695 | 0.488 |
| | YES | 104 (34.3%) | 46.25±9.69 | | |
| Demand Accompanied by Family Members | NO | 72 (23.8%) | 44.04±9.40 | -1.649 | 0.100 |
| | YES | 231 (76.2%) | 46.23±9.94 | | |

*, $P<0.05$, indicating that the data is statistically significant, and age, educational level, occupation, place of residence, per capita monthly income and the number of pregnancies are the influencing factors of anxiety; F, single factor analysis of variance; t, independent sample T test.

pregnant women with fetal magnetic resonance examination. Therefore, to explore the results of this study, we mainly refer to the psychological intervention study of pregnant women in the process of magnetic resonance imaging and the psychological study of pregnant women in prenatal examination.

**Table 2. Logistic regression analysis of the influencing factors of anxiety in pregnant women undergoing fetal magnetic resonance examination.**

| Variable | Coefficient | SE | P value | OR | 95%CI |
|---|---|---|---|---|---|
| Constant | -2.764 | 0.847 | 0.001 | 0.063 | — |
| college degree | 0.782 | 0.342 | 0.022 | 2.186 | (1.119,4.273) |
| country | 0.771 | 0.361 | 0.033 | 2.162 | (1.066,4.385) |

SE, standard error; OR, odd ratio; 95% CI, 95% confidence interval.

In this study, anxiety screening was performed on 303 pregnant women who underwent fetal magnetic resonance examination. The results showed that the positive rate of anxiety was 31.02% (94/303), which was higher than that of ordinary pregnant women (28.8%) [24] and higher than that of pregnant women with spontaneous abortion (11.43%) [25]. It can be seen that pregnant women with fetal magnetic resonance examination have a higher degree of anxiety, suggesting that health care personnel should pay attention to the mental health of the population, eliminate the influencing factors of anxiety, and take corresponding measures to intervene early.

The results of this study found that the education level and place of residence of pregnant women were risk factors for anxiety in pregnant women with fetal magnetic resonance examination. The results of this study found that pregnant women with low educational levels were more likely to have anxiety than pregnant women with high educational levels. In this study, patients with a low educational level had the highest anxiety score, which was consistent with the study of Yang [26]. The reason may be that pregnant women with low education levels receive less social support and have poor understanding of the relevant knowledge of pregnancy and the health knowledge of the fetus [27]. They cannot correctly understand the relevant information of magnetic resonance examination during pregnancy or mistakenly believe that magnetic resonance examination is to increase the efficiency of the hospital from the perspective of distrust, which leads to resistance [28]. In addition, some magnetic resonance examination costs are high, and pregnant women are worried that they cannot achieve the purpose of examination while bearing the economic burden, so the incidence of anxiety is high. An unsatisfactory living environment can also cause anxiety [29]. According to the research report [27], compared with cities, country residence is not convenient to obtain information and lacks social resources, which cannot meet the needs of physical and mental health and is more likely to lead to adverse emotions. In addition, pregnant women living in rural areas know little about magnetic resonance examinations during pregnancy due to differences in medical levels, and the noisy environment in the process causes uncomfortable reactions and aggravates anxiety. Therefore, for such pregnant women, it is necessary to give individualized pregnancy guidance, strengthen the education of health care and examination-related knowledge, improve the cognitive level of pregnant women, and reduce the possibility of anxiety.

There are also some limitations in this study: ①This study used a cross-sectional survey and only one research hospital was included. In the future, the sample size and multicenter cooperation can be increased. ②The influence of other factors on the mood of pregnant women during fetal magnetic resonance examination has not been studied. In the future, factors such as family, society, prenatal high-risk pregnant women and fetal abnormality diagnosis can be included for further research.

## Conclusion

In this study, the anxiety status of pregnant women undergoing fetal magnetic resonance examination was investigated to determine the positive rate of anxiety and its influencing factors. A low education level and living in a country will increase the probability of anxiety in pregnant women during magnetic resonance examination. The results of this study suggest that medical personnel should pay attention to assessing the mental health of pregnant women during routine pregnancy care and actively carry out examination-related health education to improve their mental health.

## Author Contributions

**Conceptualization:** Yun Wang, Hui Zhang.

**Data curation:** Yun Wang.

**Formal analysis:** Xue Liu.

**Investigation:** Yuping Zheng, Hui Zhang.

**Methodology:** Yuping Zheng, Juan Liu.

**Project administration:** Juan Liu, Gang Ning.

**Resources:** Xuesheng Li.

**Software:** Yuping Zheng, Yang Liu.

**Supervision:** Li Zhang, Gang Ning.

**Validation:** Xuesheng Li, Gang Ning.

**Visualization:** Gang Ning.

**Writing – original draft:** Yuping Zheng.

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
