## [Decision Letter · Decision Letter 0]

3 Sep 2023

PONE-D-23-23147Study on influencing factors of anxiety in pregnant women during the first fetal magnetic resonance examination: a cross-sectional studyPLOS ONE

Dear Dr. Wang,

Thank you for submitting your manuscript to PLOS ONE. After careful consideration, we feel that it has merit but does not fully meet PLOS ONE’s publication criteria as it currently stands. Therefore, we invite you to submit a revised version of the manuscript that addresses the points raised during the review process.

In addition to the comments from reviewers, please try to provide more context in your introduction especially explaining more what you mean by anxiety of pregnant women with possibility of fetal abnormalities. Also, proof-read your paper to refine the language. ==============================

We look forward to receiving your revised manuscript.

Kind regards,

Aloysius Gonzaga Mubuuke

Academic Editor

PLOS ONE

Additional Editor Comments:

There is need to expound on your introduction to put more context to the study especially explaining anxiety of pregnant women in relation to fetal abnormalities.

Reviewers' comments:

Reviewer's Responses to Questions

**Comments to the Author**

1. Is the manuscript technically sound, and do the data support the conclusions?

Reviewer #1: Yes

Reviewer #2: No

2. Has the statistical analysis been performed appropriately and rigorously? 

Reviewer #1: No

Reviewer #2: Yes

3. Have the authors made all data underlying the findings in their manuscript fully available?

Reviewer #1: Yes

Reviewer #2: Yes

4. Is the manuscript presented in an intelligible fashion and written in standard English?

Reviewer #1: No

Reviewer #2: Yes

5. Review Comments to the Author

Reviewer #1: Dear authors,

Congratulations on your work regarding the subject of anxiety factors in pregnant women during the first fetal magnetic resonance examination. This study is relatively novel; therefore, I highly recommend that it be revised, edited, and improved to attract the attention it deserves from readers.

Abstract

Title: In line with the study objective, the title should incorporate the anxiety positivity rate. Please include more relevant conclusions drawn from the results presented in the abstract, particularly the factors influencing anxiety. Alternatively, you should first mention the anxiety mean score and the positive rate of anxiety in the results section. Subsequently, you can elaborate on those conclusions.

Introduction

This section is quite brief. Please provide information regarding the proportion of women undergoing Fetal Magnetic Resonance examination in your country and elaborate on the psychological effects of the procedure.

Methods

The authors state, "The exclusion criteria are as follows: ① Presence of metal implants in pregnant women." Please note that this cannot serve as an exclusion criterion as these women are not included in the examination at all.

Results Please refrain from duplicating the results in both tables and the text.

You should avoid classifying anxiety into three categories because you have not described mild, moderate, and severe anxiety in the methods section.

Using multiple logistic regression analysis is more appropriate than 'a multi-factor linear regression equation' due to the binary nature of the dependent variable.

Please provide a description of what you mean by 'place of residence' in the methods section.

Discussion The first 5 lines are suitable for the introduction section: "Magnetic resonance has become an important supplementary means... and worries about fetal health [9]). You should start this section with your objectives. Afterward, mention a summary of your results, including the mean score of anxiety or the percentage of severe anxiety and associated factors. Then, compare your results with other studies.

These sentences are not appropriate in this context and should be removed and placed in the introduction section: "Pregnant women, as a special group of women, not only face the fear of the unknown examination but also worry about the impact of the examination on themselves and the fetus, as well as the possible economic burden caused by fetal diseases, so the risk of anxiety is high."

Limitations

Ok

Conclusions

Ok

Clinical Implications

Ok

References

Ok

Reviewer #2: The introduction is very short. Anxiety of pregnant women with the possibility of fetal abnormality is not well explained? What are the anxiety risk factors of these pregnant women?

The regression report should be presented in the abstract in terms of coefficients and positive and negative predictors.

The calculation of the sample size is not clear.

Were these pregnancies high-risk medically, for example, gestational hypertension, diabetes and...

Regression model, independent variables are not well introduced.

Tables should be included legend with proper explanation about the variables , models of statistics, cut off of anxiety ,......

An important question is whether the anxiety of pregnant women is caused by worrying about the result of the test and the abnormality of the fetus, or is it because of doing the magnet test?

This is a correlational study. Please avoid of causes ,......in the reporting of the data the paper. For example:

In summary, this study believes that the anxiety of pregnant women undergoing magnetic resonance examination for the first time is caused by multiple factors.........................

The recommendations and clinical application are explained in more detail.

6. PLOS authors have the option to publish the peer review history of their article (what does this mean?). If published, this will include your full peer review and any attached files.

Reviewer #1: **Yes: **Forough Mortazavi

Reviewer #2: No

---

## [Author Response · Author response to Decision Letter 0]

17 Oct 2023

Dear editors and reviewers:

We are very grateful that you and the reviewers for providing us the opportunity to revise and improve the quality of our manuscript (manuscript number: PONE-D-23-23147; manuscript entitled “Study on influencing factors of anxiety in pregnant women during the first fetal magnetic resonance examination: a cross-sectional study”.

After carefully reading the comments of the reviewers, we have extensively revised the manuscript point-by-point accordingly. In this revised version, changes to our manuscript were all highlighted within the document by using red-colored text. We believe that the quality of our manuscript has been substantially improved and respectfully hope that you and reviewers approve its publication in “PLOS ONE”. If there is still something that needs to be revised and improved, please do not hesitate to contact us, and we will revise it carefully again.

Thank you and best regards.

Yun Wang (corresponding author):

Address: Department of Radiology Nursing, West China Second University Hospital, Sichuan University, Key Laboratory of Obstetric and Gynecologic and Pediatric Diseases and Birth Defects of Ministry of Education

Email: 532410227@qq.com.

According to your modification suggestions, we reply one by one as follows:

Editorial Team

In addition to the comments from reviewers, please try to provide more context in your introduction especially explaining more what you mean by anxiety of pregnant women with possibility of fetal abnormalities. Also, proof-read your paper to refine the language.

Response:

We feel great thanks for your professional review work on our article. According to your nice suggestions, we have added relevant background in the introduction, in particular a detailed explanation of the effects of maternal anxiety on the fetus, and have revised the language of the article (Page 2 Line 16-29, Page 3 Line 1-26, in the Revised Manuscript with Track Changes). If any modification is inappropriate, please do not hesitate to tell us, and we will do our best to carefully modify according to the requirements.

Reviewer #1:

Congratulations on your work regarding the subject of anxiety factors in pregnant women during the first fetal magnetic resonance examination. This study is relatively novel; therefore, I highly recommend that it be revised, edited, and improved to attract the attention it deserves from readers.

1. Abstract

Title: In line with the study objective, the title should incorporate the anxiety positivity rate. Please include more relevant conclusions drawn from the results presented in the abstract, particularly the factors influencing anxiety. Alternatively, you should first mention the anxiety mean score and the positive rate of anxiety in the results section. Subsequently, you can elaborate on those conclusions.

Response:

We sincerely thank the reviewer for your valuable feedback that we have used to improve the quality of our manuscript. According to your nice suggestions, based on the study objectives, we added the anxiety positive rate to the title. The revised title is “Study on the positivity rate and influencing factors of anxiety in pregnant women during the first fetal magnetic resonance examination: a cross-sectional study”. At the same time, we added the positive rate of anxiety (31.02%) and the average score of anxiety (45.71±9.84) to the results of the abstract to make the content more perfect. In addition, the research results are also expounded in the conclusion (Page 1 Line 22-23, Page2 Line 5-7, in the Revised Manuscript with Track Changes). If there are any further changes that need to be made, we will be very willing to make changes as required.

2. Introduction

This section is quite brief. Please provide information regarding the proportion of women undergoing Fetal Magnetic Resonance examination in your country and elaborate on the psychological effects of the procedure.

Response:

Thank to the reviewer for the good questions. Under the guidance of your nice suggestions, we reviewed the literature and improved the content of the introduction part. Unfortunately, there is a lack of research in our country on the percentage of women who undergo fetal magnetic resonance examination. However, through other intervention studies and foreign studies, we have supplemented the psychological effects of the procedure on pregnant women (Page 2-3, in the Revised Manuscript with Track Changes). If any modification is inappropriate, please do not hesitate to tell us, and we will do our best to carefully modify according to the requirements.

3. Methods

The authors state, "The exclusion criteria are as follows: ①Presence of metal implants in pregnant women." Please note that this cannot serve as an exclusion criterion as these women are not included in the examination at all.

Response:

We are really sorry for our careless mistakes. Thank you very much for your kind reminder. Due to my misunderstanding, I mistakenly included "Presence of metal implants in pregnant women" in the exclusion criteria. According to your nice suggestion, we have removed this exclusion criterion (Page 4 Line 12-13, in the Revised Manuscript with Track Changes).

4. Results 

Please refrain from duplicating the results in both tables and the text.

You should avoid classifying anxiety into three categories because you have not described mild, moderate, and severe anxiety in the methods section.

Using multiple logistic regression analysis is more appropriate than 'a multi-factor linear regression equation' due to the binary nature of the dependent variable.

Please provide a description of what you mean by 'place of residence' in the methods section.

Response:

We feel great thanks for your professional review work on our article. According to your nice suggestions, we have made extensive corrections to our previous draft, the detailed corrections are listed below.

a. According to your nice suggestion, we have removed the duplicate results in the article (Page 6 Line 1-2, in the Revised Manuscript with Track Changes). 

b. Thank you very much for your kind reminder. We deleted the classification of anxiety levels to ensure the consistency of the article (Page 6 Line 1-2, in the Revised Manuscript with Track Changes).

c. Thanks to the reviewer for reminding me. Since We misunderstood the statistical method of multi-factor regression, We have carried out logistic regression again under your professional suggestion. The final results showed that college degree and country residence were the influencing factors of anxiety in pregnant women undergoing fetal magnetic resonance examination (Page 7 Table 2, in the Revised Manuscript with Track Changes).

d. Thank you very much for your kind reminder. We have described the meaning of place of residence in the methods section (Page 4 Line 20-21, in the Revised Manuscript with Track Changes).

5. Discussion The first 5 lines are suitable for the introduction section: "Magnetic resonance has become an important supplementary means... and worries about fetal health [9]). You should start this section with your objectives. Afterward, mention a summary of your results, including the mean score of anxiety or the percentage of severe anxiety and associated factors. Then, compare your results with other studies.

These sentences are not appropriate in this context and should be removed and placed in the introduction section: "Pregnant women, as a special group of women, not only face the fear of the unknown examination but also worry about the impact of the examination on themselves and the fetus, as well as the possible economic burden caused by fetal diseases, so the risk of anxiety is high."

Response:

Thanks to the reviewer for the good questions. Under your guidance, we have revised the discussion section and removed some inappropriate statements or put them in the introduction (Page 2 Line 25-29, Page 7 Line 14-19, in the Revised Manuscript with Track Changes). If any modification is inappropriate, please do not hesitate to tell us, and we will do our best to carefully modify according to the requirements.

Reviewer #2:

1. The introduction is very short. Anxiety of pregnant women with the possibility of fetal abnormality is not well explained? What are the anxiety risk factors of these pregnant women?

Response:

Thank you very much for your kind reminder. We feel great thanks for your professional review work on our article. By reviewing the literature, we supplement the introduction, which focuses on explaining the effects of maternal anxiety on the fetus. At the same time, the anxiety factors of pregnant women in fetal magnetic resonance examination are summarized. We believe that the influencing factors mainly come from the examination equipment and the subjects themselves according to the literature, and explain the two factors in detail. If there is still a need to improve the part, please do not hesitate to contact us, we will carefully modify (Page 2 Line 16-29, Page 3 Line 8-26, in the Revised Manuscript with Track Changes).

2. The regression report should be presented in the abstract in terms of coefficients and positive and negative predictors.

Response:

Thanks to the reviewer for the good questions. We have re-improved the expression of the regression results and displayed them in the abstract (Page 1 Line 27-28, in the Revised Manuscript with Track Changes). 

3. The calculation of the sample size is not clear.

Response:

Thank you very much for your kind reminder. This study is based on the calculation method of sample size in the Chinese Journal of Nursing. The sample size should be at least 5-10 times the number of variables. Considering that there were 20% invalid samples, the final sample size of this study was 84-168 cases (Page 4 Line 13-17, in the Revised Manuscript with Track Changes). If there are any other modifications we could make, we would like very much to modify them and we really appreciate your help.

4. Were these pregnancies high-risk medically, for example, gestational hypertension, diabetes and...

Response:

Thanks to the reviewer for the good questions. According to your nice suggestion, we have conducted a literature review. High-risk pregnancy refers to the occurrence of complications or certain pathogenic factors that may endanger the life safety of pregnant women, fetuses and newborns during pregnancy and require early intervention or related treatment, including: (1) Pregnant women younger than 18 years old or older than 35 years old. (2) Have a history of abnormal pregnancy, such as abortion, premature birth, etc. (3) Various pregnancy complications, such as eclampsia. (4) Various pregnancy comorbidity, such as heart disease. (5) May occur abnormal childbirth. (6) Placental insufficiency. (7) Other pathological obstetric problems and abnormal pregnancy, etc. Therefore, gestational hypertension and diabetes are clinically high-risk.

Reference: 

Lin Yi. Current situation, problems and directions of high-risk pregnancy research in China. Journal of Shanghai Jiaotong University (Medical Science). 2022; 42(04):403-408.

Link:https://kns.cnki.net/kcms/detail/detail.aspx?FileName=SHEY202204001&DbName=CJFQ2022

5. Regression model, independent variables are not well introduced.

Response:

Thanks to the reviewer for the good questions. Under your nice guidance, we have revised the introduction description of independent variables and described all dummy variables as references (Page 7 Line 5-11, in the Revised Manuscript with Track Changes). If there are any other modifications we could make, we would like very much to modify them and we really appreciate your help.

6. Tables should be included legend with proper explanation about the variables, models of statistics, cut off of anxiety......

Response:

Thank you very much for your kind reminder. We have supplemented the explanations of tabular variables and statistical models (Page 6 Table 1, Page 7 Table 2, in the Revised Manuscript with Track Changes).

7. An important question is whether the anxiety of pregnant women is caused by worrying about the result of the test and the abnormality of the fetus, or is it because of doing the magnet test?

Response:

We feel great thanks for your professional review work on our article. Through literature review, it can be seen that the anxiety of pregnant women in this study mainly comes from the examination equipment and the pregnant women themselves. The anxiety of the pregnant women themselves includes concerns about the results of prenatal diagnosis and the impact of the examination on the fetus. However, such factors are not explored in detail in this study, which is also the limitation of this study. In the future, our team will learn from the experience and incorporate the relevant factors of pregnant women's family, social and prenatal diagnosis to carry out more comprehensive research. Thank you again for your valuable suggestions on the article.

8. This is a correlational study. Please avoid of causes ,......in the reporting of the data the paper. For example:

In summary, this study believes that the anxiety of pregnant women undergoing magnetic resonance examination for the first time is caused by multiple factors.........................

The recommendations and clinical application are explained in more detail.

Response:

Thanks to the reviewer for the good questions. We removed some inappropriate statements and used accurate language to explain correlation studies (Page9 Line12-16, in the Revised Manuscript with Track Changes). If there is any part of the expression that is not perfect, we will continue to revise it carefully.

---

## [Decision Letter · Decision Letter 1]

30 Oct 2023

PONE-D-23-23147R1Study on the positivity rate and influencing factors of anxiety in pregnant women during the first fetal magnetic resonance examination: a cross-sectional studyPLOS ONE

Dear Dr. Wang,

Thank you for submitting your manuscript to PLOS ONE. After careful consideration, we feel that it has merit but does not fully meet PLOS ONE’s publication criteria as it currently stands. Therefore, we invite you to submit a revised version of the manuscript that addresses the points raised during the review process.

The paper has greatly improved. Please attend to the minor comments from one of the reviewers. In addition, proof-read the entire paper to correct the occassional language errors.==============================

We look forward to receiving your revised manuscript.

Kind regards,

Aloysius Gonzaga Mubuuke

Academic Editor

PLOS ONE

Journal Requirements:

Additional Editor Comments:

One of the reviewers requests to attend to some minor issues to further strengthen the paper. In addition, please proof-read the entire paper to correct the occasional grammatical errors.

Reviewers' comments:

Reviewer's Responses to Questions

**Comments to the Author**

1. If the authors have adequately addressed your comments raised in a previous round of review and you feel that this manuscript is now acceptable for publication, you may indicate that here to bypass the “Comments to the Author” section, enter your conflict of interest statement in the “Confidential to Editor” section, and submit your "Accept" recommendation.

Reviewer #1: All comments have been addressed

Reviewer #2: All comments have been addressed

2. Is the manuscript technically sound, and do the data support the conclusions?

Reviewer #1: Yes

Reviewer #2: Yes

3. Has the statistical analysis been performed appropriately and rigorously? 

Reviewer #1: Yes

Reviewer #2: Yes

4. Have the authors made all data underlying the findings in their manuscript fully available?

Reviewer #1: Yes

Reviewer #2: Yes

5. Is the manuscript presented in an intelligible fashion and written in standard English?

Reviewer #1: No

Reviewer #2: Yes

6. Review Comments to the Author

Reviewer #1: Dear authors,

Thank you for revising the manuscript. The Introduction section has been improved; however, there is a lack of elaboration on the psychological effects of the fetal magnetic resonance examination procedure based on previous studies. The current content discusses the effects of chronic anxiety during pregnancy on fetal development. In my opinion, the manuscript requires further editing.

Reviewer #2: The authors addressed well the comments. I Thank you for following the suggestions and improving the manuscript.

7. PLOS authors have the option to publish the peer review history of their article (what does this mean?). If published, this will include your full peer review and any attached files.

Reviewer #1: **Yes: **Forough Mortazavi

Reviewer #2: No

---

## [Author Response · Author response to Decision Letter 1]

30 Nov 2023

Dear editors and reviewers:

We are very grateful that you and the reviewers for providing us the opportunity to revise and improve the quality of our manuscript (manuscript number: PONE-D-23-23147R1; manuscript entitled “Study on the positivity rate and influencing factors of anxiety in pregnant women during the first fetal magnetic resonance examination: a cross-sectional study”.

According to your nice suggestions, we have extensively revised the manuscript point-by-point accordingly. In this revised version, changes to our manuscript were all highlighted within the document by using red-colored text. We believe that the quality of our manuscript has been substantially improved and respectfully hope that you and reviewers approve its publication in “PLOS ONE”. If any modification is inappropriate, please do not hesitate to tell us, and we will do our best to carefully modify according to the requirements.

Thank you and best regards.

Yun Wang (corresponding author):

Address: Department of Radiology Nursing, West China Second University Hospital, Sichuan University, Key Laboratory of Obstetric and Gynecologic and Pediatric Diseases and Birth Defects of Ministry of Education

Email: 532410227@qq.com.

According to your nice modification suggestions, we reply one by one as follows:

Editorial Team

The paper has greatly improved. Please attend to the minor comments from one of the reviewers. In addition, proof-read the entire paper to correct the occassional language errors.

One of the reviewers requests to attend to some minor issues to further strengthen the paper. In addition, please proof-read the entire paper to correct the occasional grammatical errors

Response:

We sincerely thank you for your careful reading. Based on your nice suggestions, we tried our best to modify the language expression and language errors in the manuscript and marked them in red, but these changes will not influence the content and framework of the manuscript. In addition, regarding the revision suggestions of reviewers, we have supplemented the corresponding contents by consulting literature (Page 2 Line 22-29, Page 3 Line 1-4 in the Revised Manuscript with Track Changes). We appreciate for your warm work earnestly and hope that the correction will meet with approval.

Reviewer #1:

Dear authors,

Thank you for revising the manuscript. The Introduction section has been improved; however, there is a lack of elaboration on the psychological effects of the fetal magnetic resonance examination procedure based on previous studies. The current content discusses the effects of chronic anxiety during pregnancy on fetal development. In my opinion, the manuscript requires further editing.

Response:

Thank you very much for your kind reminder. According to your nice and professional suggestions, we have supplemented the elaboration of the related psychological effects of fetal magnetic resonance examination by consulting literature, and highlighted the importance of anxiety in fetal magnetic resonance examination in order to better link up the content of the article (Page 2 Line 22-29, Page 3 Line 1-4 in the Revised Manuscript with Track Changes). If there are any other modifications we could make, we would like very much to modify them and we really appreciate your help.

---

## [Editor Report · Decision Letter 2]

2 Jan 2024

Study on the positivity rate and influencing factors of anxiety in pregnant women during the first fetal magnetic resonance examination: a cross-sectional study

PONE-D-23-23147R2

Dear Dr. Wang,

We’re pleased to inform you that your manuscript has been judged scientifically suitable for publication and will be formally accepted for publication once it meets all outstanding technical requirements.

Kind regards,

Aloysius Gonzaga Mubuuke

Academic Editor

PLOS ONE

Additional Editor Comments (optional):

All comments have been addressed
---

## [Editor Report · Acceptance letter]

10 Jan 2024

PONE-D-23-23147R2 

PLOS ONE

Dear Dr. Wang, 

I'm pleased to inform you that your manuscript has been deemed suitable for publication in PLOS ONE. Congratulations! Your manuscript is now being handed over to our production team.

Kind regards, 

on behalf of

Dr. Aloysius Gonzaga Mubuuke 

Academic Editor

PLOS ONE